# Postharvest Water Loss of Wine Grape: When, What and Why

**DOI:** 10.3390/metabo11050318

**Published:** 2021-05-14

**Authors:** Chiara Sanmartin, Margherita Modesti, Francesca Venturi, Stefano Brizzolara, Fabio Mencarelli, Andrea Bellincontro

**Affiliations:** 1Department of Agriculture, Food and Environment, University of Pisa, via del Borghetto 80, 56124 Pisa, Italy; chiara.sanmartin@unipi.it (C.S.); francesca.venturi@unipi.it (F.V.); fabio.mencarelli@unipi.it (F.M.); 2Interdepartmental Research Center, Nutraceuticals and Food for Health, University of Pisa, Via del Borghetto 80, 56124 Pisa, Italy; 3Institute of Life Sciences, Scuola Superiore Sant’Anna, Piazza Martiri della Libertà 33, 56127 Pisa, Italy; margherita.modesti@santannapisa.it; 4Department for Innovation in Biological, Agro-Food and Forest Systems, University of Tuscia, Via S. Camillo de Lellis, 01100 Viterbo, Italy

**Keywords:** dehydration, withering, wine grape, volatile organic compounds (VOCs), polyphenols, anthocyanins, sensory profile, non-destructive monitoring

## Abstract

In postharvest science, water loss is always considered a negative factor threatening fruit and vegetable quality, but in the wine field, this physical process is employed to provide high-quality wine, such as Amarone and Passito wines. The main reason for this is the significant metabolic changes occurring during wine grape water loss, changes that are highly dependent on the specific water loss rate and level, as well as the ambient conditions under which grapes are kept to achieve dehydration. In this review, hints on the main techniques used to induce postharvest wine grape water loss and information on the most important metabolic changes occurring in grape berries during water loss are reported. The quality of wines produced from dried/dehydrated/withered grapes is also discussed, together with an update on the application of innovative non-destructive techniques in the wine sector. A wide survey of the scientific papers published all over the world on the topic has been carried out.

## 1. Introduction

The value chain of a wine coming from postharvest grape water loss is significantly higher than for a regular wine without this postharvest step. This is linked to the fact that water loss is a dramatic event that can improve or worsen the final wine quality. In ancient times, and still currently in some geographical areas, this physical process is naturally occurring, with a strong effect being the specific climatic conditions at which the harvested grape bunches are placed in order to lose water. However, scientific and technological evolution have shown that the adoption of simple or more complex technologies in order to guarantee the proper water loss from grape berries is extremely useful for the final wine quality and does not compromise berry aromatic and chemical features. In this review the postharvest protocols used as practice or technique, together with the main biochemical changes induced by these practices on grape and wines, as well as the application of non-destructive sensors to monitor the process are, dealt with.

## 2. Postharvest Techniques to Induce Water Loss

Mencarelli and Tonutti (2013) [1] defined the terminology (glossary) used widely to indicate the same physical process: water loss from grape berries. Dehydration, drying, withering, and raisining are different terms to indicate the same process. However, each term refers to the specific applied practice or technique [2]. A practice is an activity corresponding to the faculties conferred by custom in the specific context of performance or behavior (custom or ritual custom), while a technique is any activity that, on the basis of scientific knowledge, designs instruments, appliances, machines, motors, or other tools intended to meet the practical needs of life. Starting from these definitions, we can classify the systems used to remove water from grape berries as practices or techniques (Table 1).

Not all the practices/techniques reported in Table 1 are operated in postharvest conditions. Can-cut, late harvest, noble rot, and ice wine practices have been included because they are used to induce water loss and to produce sweet wine, even though they are not carried out in postharvest. Below, we only discuss postharvest practices/techniques.

### 2.1. Sun-Drying

This is the ancestral practice of drying food in countries where sun is available most of the year, such as in the Mediterranean Basin area. The sun drying practice was initially used to make raisins as food and only later used for winemaking. Most likely, the first wine produced came from partially dried grapes that were left in a clay jar or pot and started to ferment [3]. Why must this system, which is used for the most famous sweet Passito wines produced in the Mediterranean Basin, such as Commandaria, Santorini Vin Santo, Passito di Pantelleria, and Pedro Ximenez, be considered a practice? The reason is that the water loss of berry is promoted by the natural conditions where berries are left to dry, with sun and wind, thus, low relative humidity (RH), being the main drivers that favor the water release from the berries. In this context, no human action to control the environment is assumed. Water loss is triggered mainly by heat (sun) and wind (which removes the boundary layer allowing for a continuous vapor pressure deficit). Both these environmental factors maintain the ambient RH low, even though the area where this practice is used is, in most cases, very close to the sea. In the classification made by Mencarelli and Bellincontro (2013) [4], this practice is listed as very fast or fast process (from 5 days, in Andalusia, to 2–3 weeks, in some areas of Sicily). The other characteristic that must be taken into account is the level of water loss, which is usually higher than 30%. The typical wine characteristics are high sweetness; low acidity; high volatile acidity (depending on the process speed; the faster the process, the lower the amount); and few but very well characterized wine aromatic traits, such as honey, dry figs, dry plums, and caramel. The wines from this practice are easily identifiable and very often the balance between acetic acid and sugar content must be managed during vinification [5]. Rarely are the grape aromatic compounds recognizable unless aromatic varieties are used, such as Muscat type.

### 2.2. Sun-Drying in Plastic Tunnel

The sun-drying practice has been technologized by using transparent plastic film tunnels. The harvested grape bunches are placed on plastic mats or in perforated plastic crates on the ground in a single layer, and the extremities of 10–50 m long tunnels are left open to create an air stream, which can be accelerated by using floor fans. The reason plastic films are employed is to protect grape bunches from rain and dew. The dehydration process is much faster because the temperature of the air inside the tunnel reaches high levels (e.g., 50 °C), compromising the pool of wine aromatic compounds. It is known that the employment of high temperatures for grape drying strongly contributes to VOC oxidation, favoring the production of oxidized compounds, such as furfurals or lactones [6].

### 2.3. Uncontrolled Dehydration in Fruttaio (Closed Facility) 

This is an ancient practice used in countryside areas (mainly at high latitude) where the ambient conditions do not permit a proper grape drying in open fields under the sun. Grape bunches can be placed in plastic crates one over the other to create a high stack or, as traditionally done, on a straw mat or hanging from a vertical metallic net. Whatever the system of grape bunch placement, the practice is held in closed facilities and, traditionally, is based on natural ventilation coming from opened windows. Today, ventilation is fostered by using floor fans. This practice/technique relies on the outer environmental conditions, which should be dry regardless of the temperature. If the temperature is high, the process is faster, while if the temperature is low, the aromatic quality of grape berries are better maintained. It has been demonstrated that the extreme variation of temperature and RH in these facilities alters the grape aroma panorama compared to facilities where the ambient conditions are controlled, favoring an activation of lipoxygenase and alcohol dehydrogenase enzymes [7,8].

### 2.4. Controlled Dehydration in Fruttaio (Closed Facility)

The difference with the aforementioned practice is that, in this case, the dehydration facility allows a complete control of the ambient conditions with cooling/heating tools and a dehumidifying plant beyond the ventilation. The temperature and the RH can be fixed, and thus, the dehydration time can be programmed. These facilities guarantee, depending on the size of the facility and the efficiency of the plant, the health of the bunches, significantly reducing the losses that, on the other hand, characterize the aforementioned practice. The use of low-temperature dehydration allows for excellent results in terms of phenolic and aromatic compounds, preserving the VOC profile of grape varieties and reducing the formation of oxidized compound and volatile acidity [9,10]. The control of dehydration conditions applying this specific drying approach allow the carrying out of reliable studies on berry metabolism during wine grape water loss [11], allowing us to have a clearer idea of the effect of water loss level and rate on berry metabolism [12,13].

### 2.5. Withering in Fruttaio with Ambient Control

This is the common technique used in Valpolicella (Italy) to produce grapes for Amarone wine. It is known as an open–closed system because the facility used for withering (neither drying nor dehydration) has a cooling and dehumidifying plant with ventilation but also an automatic system to open the windows when the outside environmental conditions are suitable. The main characteristic of this technique from a water loss viewpoint is that this physical process takes a long time under such conditions. Indeed, the production Disciplinary of Amarone wine reports that the withered berries must be processed after the 1st of December, meaning more than two months from the harvest of the main employed varieties. The long term withering accounts for the quality of grape berries that undergo a mild water stress and senescence process, triggering the overexpression/repression of genes involved in the different biochemical pathways responsible for the beneficial metabolic changes that occur running a proper berry dehydration [11,14]. In fact, significant changes of VOCs occur in grape berries of Corvina, Corvinone, and Rondinella, the three main varieties of Amarone wine, depending on the amount of water loss [15] affecting significantly the final wine characteristics [16].

## 3. Metabolite Changes in Wine Grape during Postharvest Water Loss

The most crucial aspect of grape dehydration is tissue water loss and the consequent weight loss of berries. Weight loss can largely vary in terms of percentage, from relatively low values (20–30%) for dehydrated grapes destined to produce dry wines, to high levels (40–50%) in the case of sweet wines, with the consequent concentration of sugars and extractives in general [4,17]. The dehydration causes deep chemical–physical modifications of the grapes as a function of grape features (variety, size of the berry, compactness of the bunch, berry skin surface, micropores and cracks, etc.) but also of the environmental conditions (ventilation, relative humidity of the air, temperature, air flow, sunlight) and of the process length [1,18,19]. As far as grape berry metabolites are concerned, it is very important to consider how to express their concentration. If the objective is to study the evolution of berry composition during postharvest dehydration, it can be more appropriate to express the metabolite concentration per single berry or with respect to dry weight in order to avoid bias due to the concentration effect owing to the water loss.

Postharvest water loss induces a plethora of different metabolic changes in grape berries and, consequently, in the final wine composition.

Among the quality aspects that have been investigated in recent years, most of the published studies focused on compositional changes affecting wine color, phenolic profile, and aroma compounds, together with an evident increase of sugar content, mainly due to a concentration effect [2,7,20,21,22,23,24,25,26,27,28]. These are, indeed, the quality traits that are mainly affected by water loss and its metabolic consequences, and they are also the most important for both consumer’s acceptance and nutraceutical value of the final product. The dramatic changes of grape berry cell structure taking place during dehydration are freeing non-volatile VOC precursors (glycosylated compounds) that, during grape crushing when enzymes are released from the cells, are metabolized and transformed into volatile compounds. The degree of water loss plays a crucial role in the concentration of these precursors. Cell structure alterations associated with grape water loss also induce enzymatic and non-enzymatic reactions that lead, in turn, to the biosynthesis/modification of important other non-volatile metabolites, such as polyphenols and anthocyanins, but also organic acids (e.g., mainly malic and tartaric acid). These events are coupled to cell response to the imposed biotic and abiotic stresses that are typically associated with water loss processes (e.g., low or high temperature, oxidation, microorganism attack), contributing to the final wine composition.

### 3.1. VOCs

Considering total VOCs synthesis/accumulation, published papers report an increase in total VOCs concentration, but also, a decrease or no significant differences have been observed when dehydration protocols are concerned, both considering free and glycosylated VOC forms [2,25,29]. 

Even if a concentration effect is normally observed for all metabolites present in the berries (e.g., sugars) due to water loss, total VOCs concentration can show different trends since it is strictly associated with the trend of several dozens of specific known molecules, which may have very different concentrations in ripened grape berries and show different behaviors in response to water loss and water stress. Water evaporation concentrates grape berry metabolites, but also active synthesis ex-novo occurs for several volatile molecules, the level of which exceeds the expected concentration effect [2,29]. However, this increase in VOCs level is cultivar-dependent and varies based on the dehydration protocol applied, potentially being linked to the specific cell sensitivity to water stress and induced enzymatic activity [30].

Grape berry dehydration can be achieved by different approaches (e.g., on- or off-vine, uncontrolled or controlled conditions, etc.), and all of them have been studied in terms of their influence on VOCs production. As an example, it has been observed that fast dehydration protocols, operated under controlled conditions, enhance VOCs emission more than long dehydration protocols, typically observed when grapes are dehydrated under natural conditions, such as the withering process [30,31].

A general effect of dehydration, which has been observed both in aromatic and non-aromatic wine grape varieties, is an increase in the synthesis/accumulation of volatile aldehydes, alcohols, esters, lactones, and terpenes, with a consequent enrichment of the resulting wine in specific compounds, especially terpenes [29,32,33].

During the early stages of grape dehydration most of the water is released from rachis, whereas in later stages, the skin represents the main route for water evaporation [4,34], and berry epicarp cells start sensing water stress producing C6 VOCs by membrane lipid oxidation [20]. One of the first events related to the wine grape metabolic response to dehydration is, indeed, the alteration of membrane permeability with the promotion of lipoxygenase (LOX) activity. This is followed by a switch from aerobic to anaerobic cell metabolism with the contribution of alcohol dehydrogenase (ADH) activity, causing the formation of specific VOCs (C6 volatiles) that provide herbaceous/green notes to grape and wine [7,20]. These compounds normally decrease with the advancement of ripening, and it has been reported that, under high dehydration levels, ADH isoenzyme activity in berry tissues increased, and that was linked to the observed general shift from aerobic to anaerobic metabolism [20,35]. This activation varies also in relation to the applied dehydration protocol [7]. Indeed, several authors observed an increase in compounds linked to the LOX metabolic pathway, such as 1-hexanol, (*E*)-2-hexen-1-ol, and (*E*)-2-hexenal, in dehydrated berries compared to control fruit [7,20,25,29], whereas other authors reported that controlled grape dehydration substantially decreased the contents of free 1-hexanol, hexanal, (*E*)-2-hexen-1-ol, and (*E*)-2-hexenal, although the effect is variety dependent [30,33,36]. Similarly, considering glycosylated C6 volatiles, published literature reports both increasing and decreasing trends for specific molecules belonging to this group [29,37]. As an example, while glycosylated 1-hexanol and (*E*)-2-hexen-1-ol increase with dehydration, (*Z*)-3-hexen-1-ol reveals an opposite behavior. These variable observations may be explained considering the very high chemical reactivity of these compounds, which tend to form other molecules, such as esters [30].

As previously mentioned, terpenes production/accumulation is also strongly affected by the dehydration process, and an increase in molecules such as geranic acid, citronellol, nerol, 3-OH-β-damascone, *cis*-furan-linalool oxide, and *cis*-pyran-linalool oxide has been observed in dehydrated grape, whereas for others, such as free linalool and geraniol, a significant decrease has been reported [2,29]. However, the accumulation trend of these compounds appears to vary largely based on cultivar and berry ripening stages. Linalool oxides are generated via linalool oxidation, which is generally triggered by water loss [38]. The observed decrease of free geraniol and linalool could be linked to molecule oxidation and conjugation. Fast dehydration protocols can be useful in order to retain the = free VOCs fraction [29]. However, tissue specific results have been described and, for instance, it has been reported that geraniol, diendiol, and *trans*-8-OH-linalool decrease in berry mesocarp during dehydration but, on the contrary, they increase in the epicarp [2,29].

The terpenoid compound’s fate is also modulated by carotenoid oxidation, which leads to the production of important VOCs, such as norisoprenoids, and can be induced by dehydration [7,31,39]. Molecules belonging to this class are, indeed, reported to be significantly affected by the dehydration process, such as vomifoliol, which increases significantly in dehydrated berries [25]. Additionally, the glycosylated terpenes’ fraction is significantly affected by dehydration, with specific compounds that have been reported to decrease, such as cis-rose oxide and hotrienol, and others which show an opposite trend, for example, geranic acid, trans-furanic-linalool oxide, nerol, linalool, citral, citronellol, and geraniol [2,29].

Among other important metabolites, the glycosylated form of methyl salicylate is also reported to increase significantly with grape dehydration [29]. Methyl salicylate is a signaling molecule that is known to affect benzenoid and terpenoid biosynthesis in fruit tissue. An increase in these two latter VOC classes has been reported in dehydrated wine grape epicarp, while the opposite trend has been observed in the mesocarp [25,40]. Volatile phenols, such as vinylguaiacol, 2,6-dimethoxyphenol, vanillin, and acetovanillone, are reported to increase as a consequence of berry dehydration. Their increasing trend could be linked to cell wall degradation via cell-wall-degrading enzyme activity, such as xylanase and endoglucanase, which are freeing non-volatile precursors for phenol biosynthesis [25,41,42,43]. 

Other important VOCs are significantly affected by the water loss rate, varying based on the applied dehydration protocol, also in relation to the grape ripening stage at harvest. Among them, ethanol, ethyl acetate, acetic acid, 2-phenylethanol, benzyl alcohol, isobutanol, isoamyl alcohol, and 1-pentanol increase with grape dehydration in several published papers, although transiently in some cases [25,29,30,43]. These compounds are indicators of an advanced fermentative process activated under dehydration, such as the oxidation of ethanol to acetaldehyde and acetic acid, which esterifies, producing ethyl acetate. Moreover, their increase may also be linked to cell death and protein/amino acid catabolism, as in the case of aromatic alcohols, such as benzyl alcohol and 2-phenylethanol, that come from the shikimic acid pathway [44,45]. Among the alcohols increasing as dehydration proceeds, ethanol can be considered a marker compound for grape dehydration [20,30]. The observed increase in free alcohols is mirrored by their glycosylated forms, which have been reported to increase in parallel to grape water loss [2,29].

Volatile esters and lactones are also significantly affected by the dehydration process. In dehydrated grape berries, a significant increase has been reported for several of these molecules, as in the case of isoamyl acetate, ethyl butanoate, and γ-valerolactone, with other compounds belonging to these classes following an opposite trend [30,43]. Results vary a lot considering different dehydration rates and specific dehydration protocols, as well as grape ripening stage at harvest. A similar variable trend has been recorded for other volatiles, such as acetoin, hexanoic, and octanoic acids, which increase until a moderate dehydration rate and decrease after reaching high water loss levels [43].

The accumulation of Maillard reaction products, such as furfural, 5-methylfurfural, and 5-hydroxymethyl-furfural, is also promoted by grape dehydration, and they contribute to toasty aromas with descriptors such as “coffee” or “chocolate” [46,47].

Overall, the significance of the compositional alterations induced by dehydration mainly depends on weight loss rates, temperature of the applied treatments, grape genotype, and ripening stage [29,30,45]. Moreover, it is important to discriminate between the effect of dehydration process on the VOCs fraction of wine grape berries and on the produced wine. Table 2 reports putative volatile markers associated to wine grape dehydration process in the cited literature.

### 3.2. Non-Volatile Metabolites

One of the most evident effects of grape dehydration is the increase in sugar content observed in dried grapes, which is substantially correlated with the achieved level of dehydration. Indeed, excluding a very early drying stage where it is possible to observe a slight increase due to a possible gluconeogenesis, reducing sugars are subjected to a negligible catabolism of glucose, with the effect of a slight decrease in the glucose-to-fructose ratio [17].

Furthermore, the grape dehydration constitutes a real biological de-acidification that can be more or less balanced by the occurring phenomena of concentration. [17]. Namely, the increase in tartaric acid, generally reported, is mainly due to the water loss and the consequent concentration effect [15,17]. Nevertheless, some authors described a decline of tartaric acid during grape drying, maybe owing to a precipitation effect as a consequence of the salification, with Ca^2+^ and K^+^ ions coming from the cell wall degradation occurring during berry drying [50]. While the increase of tartaric acid is mainly due to a concentration effect, the pattern of malic acid is more dependent on endogenous and exogenous factors. Malic acid undergoes a decline in absolute terms, and its content in dried grapes is closely related to its initial content, to the catabolic effect (malate respiration), and to the concentration effect due to grape water loss [15,17,50]. Citric acid, on the other hand, does not appear to be affected by the dehydration process [17].

Among non-volatile secondary metabolites, polyphenols and anthocyanins are reported to be significantly affected by grape dehydration processes, despite significant differences having been recorded based on different dehydration protocols and in relation to different processed cultivars [2,9,28,30,51]. With respect to this last observation, some published studies reported no significant differences in these metabolites when comparing fresh and dehydrated grapes, confirming the importance of genotype and the specific dehydration conditions, with these latter potentially inducing polyphenol and anthocyanin oxidation [9,52]. The proper temperature management in the dehydration protocol is greatly important in order to reduce phenol oxidation, assuming temperatures ranging between 10 and 20 °C as the most appropriate conditions for grape dehydration purposes [28,53,54].

The high sugar concentration observed in berry tissues could trigger the shikimic acid pathway, contributing to the increase of important classes of non-volatile metabolites [30]. It has also been reported that abiotic stresses, such as water stress, could stimulate anthocyanin and polyphenol biosynthesis due to the decrease of berry size and to the induction of ethylene biosynthesis [55,56,57], also affecting the expression of enzymes involved in the phenylpropanoid pathway [52,58]. Moreover, the activity of cell-wall-degrading enzymes can be associated to the anthocyanin induction, being related to the increase of proanthocyanidin levels, polyphenol accumulation, and liberating phenol precursors [42,43,59].

Another important event associated with the grape dehydration process is the hydrolysis of polymerized phenols that, together with a concentration effect due to water loss, could contribute to phenolic increase [22,60,61]. However, it is also important to consider that prolonged dehydration time coupled with non-optimal, or simply uncontrolled, dehydration conditions can also induce the degradation/oxidation of phenolic molecules due to the polyphenoloxidase (PPO) enzyme activity or because of some other transformation reactions [22,28,62]. These latter evidences can explain part of the variability observed comparing different published works, with some papers reporting progressive metabolite oxidation and increased PPO activity, with a consequent reduction of polyphenol content [52,63]. However, these researches assessed postharvest dehydration when phenolic ripeness already reached its peak, and at that specific time, phenols naturally tend to slowly decrease [64,65].

Specifically considering anthocyanin metabolites, a significant increase of coumaroylated compounds has been related to increasing dehydration stages [2,53]. Increased anthocyanin acylation has been associated with berry thermal stress response, indicating that this process may represent a mechanism of red pigment stabilization induced by cells to cope with thermal stress [9]. Grape pigmentation after dehydration, and consequently, wine color, varies significantly in relation to the applied dehydration protocol. As an example, traditional sun-drying approaches, which are associated to high enzymatic activity mostly referred to as oxidative reactions, is normally linked to grape and wine browning due to hydroxycinnamic acids, anthocyanins, and flavan-3-ol derivative oxidation [66].

Among specific anthocyanins significantly affected by grape dehydration, peonidin-3-*O*-glucoside, malvidin-3-*O*-caffeoylglucoside, syringetin, syringetin-3-*O*-glucoside, laricitrin-3-*O*-glucoside, and tamarixetin are reported to increase in dehydrated grapes [26]. On the other hand, the same authors reported a significant decrease of cyanidin-3-*O*-acetylglucoside, peonidin-3-*O*-acetylglucoside, prodelphinidins T1 and T2/T3, malvidin-3-*O*-acetylglucoside, petunidin-3-*O*-acetylglucoside, and delphinidin-3-*O*-acetylglucoside. These latter findings could be related in part to the higher sensitivity to oxidation of the o-diphenol aglycones [67], while the increase of acylated anthocyanins is in agreement with other published results [68]. However, cultivar specific trends have been highlighted, making less straightforward anthocyanin fate interpretation under dehydration [26,69].

As far as specific phenols are concerned, an increase in flavonol hydroxybenzoic and hydroxycinnamic acid content has often been observed in dehydrated grapes. More specifically, hydroxybenzoic acid, caftaric acid, *p*-coumaric acid derivative 1, coutaric acid glucoside, protocatechuic acid, fertaric acid, procyanidins, and quercetin glucuronide are reported to increase significantly in dehydrated grapes [28,51]. Among them, the rise of several molecules appears to be variety dependent [51]. A significant decrease in several flavonols has also been recorded, as in the case of deferuloyl hexoside pentoside, caffeic acid dihexoside, quercetin glycosides, and kaempferol-7-*O*-glucoside [26,28].

An increase in specific phenol classes has also been associated with the specific treatment applied [28]. As an example, a generalized significant enrichment in flavonols has been attributed to the increase of temperature and relative humidity within close-air type systems used in low greenhouse dehydration protocols [23,53,70,71,72]. Flavan-3-ols (condensed tannins/proanthocyanidins), which are of great importance for the final wine quality, conferring astringent and bitter properties and also contributing to color stabilization [73], generally increase in dehydrated grapes. However, their specific trend varies based on the applied dehydration conditions [28]. These observations imply the presence of different mechanisms regulating phenol composition during dehydration, with phenol groups behaving differently [74]. Also flavonols, such as astilbin, and lignans, such as isolariciresinol-β-4′-*O*-glucopyranoside, increased under dehydration [28]. However, genotype and grape growing conditions are known to greatly affect their concentration [75].

Stilbenes are important phytoalexins that increase in dehydrated grapes, such as in the case of *trans*-resveratrol, resveratrol dimer 2, resveratrol tetramer 1 and 2, *trans*-piceid, piceatannol pallidol, pallidol glucoside, (*E*)-ε-viniferin, (*E*)-astringin, and (*E*)-miyabenol C [26,28,53,61]. Conversely, several specific stilbenes are reported to decrease significantly under the dehydration process, such as (*Z*)-ε-viniferin and *cis*-piceid. The fact that different isomers follow opposite accumulation trends in dehydrated grapes has been associated to the possible employment of the (*E*)/(*Z*) ε-viniferin ratio as a withering marker [26]. The increase of these molecules under dehydration has been associated with a defense strategy, also being potentially linked to the response to high temperature employed in several protocols [53,76,77], as well as to biotic stress, such as in the case of *Botrytis cinerea, Plasmopara viticola, Erysiphe necator, Rhizopus stolonifera,* and *Aspergillus* sp. attacks [78]. Indeed, for some grape cultivars, the increase in stilbenes is considered a hallmark of natural dehydration [11,14,79]. However, stilbenes biosynthesis also quickly declines at high water loss levels due to cell corruption and consequent enzymatic and non-enzymatic oxidation, with the oligomerization state of these molecules playing a crucial role [27].

Table 3 reports putative non-volatile markers associated with the wine grape dehydration process in the cited literature.

## 4. Quality of Wine from Postharvest Dehydrated Grape

The deep chemical–physical modifications of the grapes induced by postharvest dehydration influence, directly or indirectly, the characteristics of the wine obtained, which typically exhibits a complex flavor and a strong body due to the richness in extractive substances and glycerol, which also increases the wine mellowness [80,81]. Grape dehydration allows for very dark musts with a high content of sugars and aromatic substances [18,24,82]. The dark color of must is related to the development of colored compounds known as melanoidins, coming from both non-enzymatic (i.e., Maillard reaction) and enzymatic browning reactions (i.e., polyphenoloxidase ) [83]. The formation of the primary products of Maillard reaction, 5-hydroxymethyl furfural and furfural, resulting from thermal degradation of reducing sugars, is favored by the low pH of the wine and continues during aging [84]. The browning is particularly evident when the dehydration is performed at high temperature, as observed with the direct sun exposition, as the grapes reach temperatures up to 50–55°C, compatible with the development of melanoidins [85].

During grape dehydration, researchers observed the modification of the cell wall composition and, consequently, the alteration of the permeability of the membranes, improving the extractability of some components [79,86]. Wines obtained starting from post-harvest dehydrated grapes showed higher content of phenols than the analogous wines obtained from fresh grape [61,87,88,89]. Therefore, the contribution to the rise of these molecules is not only directly dependent on their level in the processed berries, but it is probably also linked to the degradation of internal skin cell layers occurring under dehydration, which can enhance phenol and anthocyanin extractability [60,71]. As expected, the increase of the phenolic fraction is often paired with an increase in the antioxidant activity [85,90].

Regarding the anthocyanins, responsible for the color of the wine, generally a decrease in the monomeric fraction is observed [85]. It has been suggested that this decrease could be caused by the oxidation occurring during the fermentation processes, as well as by the polymerization reaction of anthocyanins with other phenols such as flavanol and tannins [85,91]. Many authors, indeed, observed an increase in the polymeric components, resulting in an increase in the color tonality [32,85,91]. Molecules belonging to this chemical class that are more prone to oxidate, such as cyanidin, tend to be generally lower in wine produced from dehydrated grapes, while other compounds that are more resistant to oxidation, such as malvidin, tend to be higher [2,92]. Additionally, it has been demonstrated that lower losses of di-substituted anthocyanins occurred in vinifying dehydrated grapes compared to fresh grapes [2].

Due to the changing of the cellular biochemistry of the grape berry, involving also the relationships between sugars and organic acid metabolism, wines produced from dried grapes may not have a good sensory equilibrium between sweetness and acidity. Organic acid concentration generally increases as a consequence of the concentration due to dehydration, even if, especially when the loss of water is slow, a decrease of the acidity may take place as a consequence of the anaerobic metabolism of the berry, leading to malic acid degradation and tartaric acid precipitation [50,60,61]. According to Zironi and Ferrarini (1987) [93], the trend of the acid content seems to be related to the temperature adopted during the drying process: it increases in the range 45–50 °C and decreases at lower temperatures (35–40 °C). Overall, the induced modifications of the organic acid level, which take place with dehydration, strongly affect palatability, taste, and freshness of the wine.

The high concentration in sugars of the must limits the growth of yeast [81,87], which suffer not only for osmotic stress, but also for the high concentration of ethanol developed, resulting in growth difficulties and altered metabolism with the synthesis of peculiar volatile compounds [94,95,96]. In this sense, fermentation problems due to the osmotic stress could be minimized by the use of osmotolerant yeast strains, which could improve the sensory characteristic of the deriving wines [97,98]. Furthermore, the glyceropyruvic pathway is predominant during the first phases of fermentation, forming a high content of glycerol, which has an intense osmotic potential. Under anaerobic conditions, glycerol and ethanol behave as electron donors, and at high pH, enable the synthesis of medium-chain fatty acids (MCFAs) [99], which may have a further toxic effect on *Saccharomyces cerevisiae* [100], causing sluggish fermentation. Lòpez de Lerma and Reinado (2011) [97] reported that wines obtained from Tempranillo dried grapes showed an increase in volatile acidity, probably because of the osmotic stress of yeast, as well as an increase in the majority of alcohols (i.e., isoamyl alcohols, 2-phenylethanol, isobutanol) and 2-methoxy-4-vinyl phenol. Interestingly, the furfural, developed during the Maillard reaction occurring in the drying process, and the 2,3-butanedione decreased significantly as a consequence of the must fermentation [97]. Chkaiban et al. (2007) [7] reported the presence of ethyl acetate, an ester associated with fruity scents, which was formed starting from the acetic acid accumulated as a consequence of the hyperosmotic stress. Moreover, in the presence of acetic bacteria, heterofermentative lactic acid bacteria, or latent fungi, a higher content of acetic acid, and therefore ethyl acetate, was observed. Another concern related to the dehydration procedure is the infection of fungi, such as *Botrytis cinerea* in the stage of gray rot, *Penicillium*, or *Aspergillus*, which must be considered as they can induce deep chemical and enzymatic changes, generating serious sensory defects in wine and synthesizing mycotoxins potentially dangerous for health [31,101,102,103].

Moreover, considering the high residual sugar often present in wine obtained from dehydrated grapes, as well as its low acidity, a microbiological stabilization (i.e., sulphitation, sterile microfiltration, etc.) is strongly recommended before bottling to avoid fermentation re-start and consequent alteration of the wine quality [104].

The aromatic profile of the final product is determined not only by the volatiles produced during the fermentation process but also by the ones coming from the grapes and developed during the dehydration [35,97]. Different chemical classes of VOCs (i.e., ketones, volatile acids, terpenes, alcohols, esters, aldehydes, etc.) have been detected in wine from postharvest dehydrated grapes as a function of water loss technique, grape variety, pedoclimatic conditions, winemaking practices, and fermentation, but the overall knowledge of the mechanism for wine flavor development is still lacking. Wine contains numerous VOCs actually, but most of them are not able to create individually a sensory impact as they are at concentrations well below their sensory threshold; wine aroma is determined by the interactions among many VOCs, and its complexity depends on the concentration of particular chemicals and on the presence of other aroma compounds, which can act synergistically or antagonistically, modifying or adding more aroma nuances [105].

The production and release of substances associated with the sensory quality is strongly related to the water-loss technology, and in this context, sunlight, storage time, and temperature play a fundamental role [98].

Slaghenaufi et al. (2020) [87] evaluated the effect of the grape withering (on-vine vs. off-vine in *fruttaio*) on the aromatic profile of Corvina wines. The wine volatile profiles were characterized by varietal compounds such as norisoprenoids, C6 alcohols, and terpenes, but also acids, esters, and higher alcohols, coming from both the withering process and fermentation. Interestingly, wine obtained by postharvest dehydration in traditional *fruttaio* were characterized by higher content of ethyl acetate, ethyl butanoate, β-citronellol, and 3-oxo-α-ionol, while β-damascenone increased in wines obtained by withering on-vine. Moreover, the evolution of some VOCs, such as linalool, (*E*)-1-(2,3,6-trimethylphenyl)-buta-1,3-diene, 3-oxo-α-ionol, ethyl acetate, n-hexyl acetate, ethyl 3-methylbutanoate, and β-damascenone, during aging was also influenced by the withering technique adopted.

High temperature and rapid dehydration induce very stressful metabolisms and chemical reactions in the grape berry, such as oxidation, amino acid catabolism (occurring due to the rapid cell death), and the Maillard reaction [10]. Consequently, a significant loss of volatiles, such as acids, esters, alcohols, terpenes, sesquiterpenes, benzene derivatives, and C6 compounds, as well as an increase in norisoprenoids and derivative compounds of furan, pyran, and lactones from the browning reactions are observed [10]. The varietal aromas are oxidized, and as a result, the wines obtained are generally characterized by a typical sensory profile where the scents of chocolate and coffee stand out, together with honey, raisin, dried apricot, and dried fig notes [10,106]. The sweet Fiano wine, obtained from grapes that are dried and botrytized, lost the varietal typical descriptors like fruity (banana, apple, pear, and pineapple), flower (lime, rose, and acacia) and vegetable notes (mint, grass, and wild fennel) to the advantage of citrus jam, dried fruits, honey, and coconut [107].

Moreno et al. (2008) [32] analyzed the volatile profile of Pinot noir wine obtained from withered grapes using a dehydration tunnel under controlled conditions (T = 22 °C; RH = 38%) in comparison with wine from fresh grapes. They observed an increase, beyond that expected by simple concentration, in many VOCs associated with floral and fruity characters (i.e., eugenol, guaiacol, citronellol, and geraniol); thus, they supposed that synthesis of these compounds occurs after harvest. Dehydration also increased the content of the norisoprenoids β-ionone, associated with the scent of raspberry, dry fruit, and violet, and β-damascenone, which has a fruity, floral, honey, berry-like aroma.

Ossola et al. (2017) [2] performed an interesting comparison between wines produced from fresh (fortified wine), partially dehydrated (Sfursat wine), and withered (passito wine) grapes of *Vitis vinifera* cv Moscato nero under controlled conditions (T = 16–18 °C; RH = 55–70%; air speed=0.6 m/s). The aromatic profiles of the wines were deeply influenced by the postharvest treatment of grapes; thus, among the VOCs identified, nerol, citronellol, linalool, and geraniol rose significantly from fortified wines compared to Sfursat and passito wines in accordance with the dehydration level of the grapes. Fortified wine exhibited an appreciably lower citral content than Sfursat and passito wines and a greater content of geranic acid. Panceri et al. (2017) [13] evaluated the effect of drying under controlled conditions (T = 7 °C, RH = 35%; air delivery capacity = 12 m^3^/s) on Merlot and Cabernet Sauvignon wines. They observed a higher content of vanillin derivatives, aldehydes, and fatty acids, as well as a high concentration of furfural in wines from dehydrated grapes, while alcohols, esters, and lactones were in higher concentration in the corresponding wines obtained immediately after harvest.

According to Mencarelli and Bellincontro (2018) [10], using low temperatures allows for preservation of the varietal VOCs; in particular, temperatures from 10 to 30 °C induce VOCs oxidation and loss in the varietal volatile compounds, even if aroma complexity is reached at 20 °C. Temperatures higher than 30 °C induce strong oxidation and tend to flatten down the peculiar aromatic profile of all the varieties.

In accordance with these considerations, it is evident that aromatic profile and volatile composition of the raw materials change as a function of grape variety and origin, dehydration techniques adopted, winemaking practices, etc. [98]. Nevertheless the main descriptors suitable to evaluate the sensory properties of wines from dehydrated grape are generally floral, thyme flower, tropical fruit such as mango and passion fruit, citrus, orange peel, apricot, dried apricot, peach, marmalade, honey, and caramel, but also black pepper and berry flavors with varietal nuances [98,107,108]. Considering in-mouth sensation, the grape dehydration generally induces a significant increase of sweetness, which is negatively correlated with acidity and bitterness. In general, due to the high alcoholic degree, the pseudo-calorific sensation, as well as the perceived bitterness, could be increased, while the high level of glycerol contributes to volume in the mouth and provides a smooth mouth sensation.

Due to the richness in aromas of wines from dehydrated grapes, aging enhances their overall sensory quality, since wood barrel VOCs (e.g., whisky lactone, eugenol, vanillin) do not create a disharmony, but rather, increase the aroma complexity with woody, spicy, and toasted notes [104]. Moreover, owing to the interaction between tannins and proteinaceous colloids, during wood maturation, a further stabilization occurs, reducing the use of fining additives necessary to reduce the turbidity but sometimes detrimental for the aromatic profile [104].

## 5. Non-Destructive Technology to Monitor Grape Postharvest Metabolic Change

As we already well clarified in the previous sections of the present manuscript, the grape dehydration process is characteristically influenced by the environmental conditions which, in turn, play a dominant role in influencing grape weight loss. Considering how most significant metabolisms and biochemical modifications are induced into the grape berry by the fruit water loss evolution and by the conditions under which this water stress occurs, it should be considered how strongly recommended the identification of powerful methods is for the process monitoring. The specific monitoring can be related to the simple observation of water loss along the process or, more in detail, referred to the possible modifications in charge to the principal metabolites (e.g., sugars, organic acids, VOCs, and polyphenols) playing a role as markers of the process. As usual for wine grape, must and wine analyses, destructive or laboratory measurements are the most recurrent, even if, commercially speaking, they can be considered expensive and, especially, time consuming [23]. Otherwise, non-destructive analytical techniques, already developed and used on fruits and vegetables in other scientific and commercial contexts including vines and wine grapes, can represent useful and alternative tools. Non-destructive technologies include fast, cost-effective non-invasive instruments for detection and monitoring of fruit quality, and their use has been pushed more and more during the last 3–4 decades [109]. Employing non-destructive technologies, analytical measurements are performed by the use of sensors based on physico-chemical properties (e.g., light energy, irradiance, fluorescence, optic, acoustic, etc.) and their interaction with the organic molecules. Chemometry is the multivariate statistical approach that allows us to combine non-destructive information to destructive attributes (chemicals) of fruit and vegetables, specifically grapes, with the goal of developing predicting models able to measure chemical parameters in intact and unknown samples [110]. In the following paragraphs the use of several non-destructive techniques aimed at measuring and monitoring grape drying processes will be discussed. In detail, some experiences based on the application of near infrared spectroscopy (NIRs), electronic nose (E-Nose), chlorophyll fluorescence, and nuclear magnetic resonance (NMR) will be cited and described as cases of study.

### 5.1. NIRs

NIRs is an optical technology based on the spectral response of the electromagnetic energy (light) into the typical NIR portion (from 780 to 2500 nm) when it interacts with organic molecules typically constituting the organic matter. When the light source emitted in the NIR wavelengths penetrates the matrices, their spectral attributes are changed because of the absorbing properties of the molecules over the scattering effect of the radiation. The main modifications due to the molecule characteristics are associated with the presence of functional groups (e.g., OH, CH, and NH), and their typical behavior in energy absorption is described as vibrational trend [111,112,113].

Bellincontro et al. (2011) [114] tested an acousto-optical tunable filter (AOTF)-NIR device, for monitoring the withering of wine grapes var. Aleatico. Usually, these grapes are dehydrated to produce sweet Passito wine, and in their work, the authors performed a controlled dehydration process in a small-scale thermo-conditioned tunnel. The followed dehydration protocol was carried out at a temperature of 20 °C, 45% of RH, and 1.5 m/s of air flow. Grape berries were dehydrated reaching 40% of weight loss and, starting from 18.7° Brix, accumulated up to a maximum sugar content of 32.4° Brix. Spectral detection was operated in reflectance by the in-contact method on the whole grape berry in the wavelength range of 1100–2300 nm, with the aim of monitoring the grape berry water loss and, concomitantly, sugar accumulation. Performed principal component analysis (PCA) models allowed them to segregate and to discriminate grouped grape samples on the basis of their progressive weight loss (0%, 5%, 10%, 15%, 20%, 25%, 30%, 35% and 40%). In that clustering, first principal component (PC1) apported the biggest contribution in terms of explained variance (97%), being referred to the water presence. In fact, as it is well known, NIR reflectance spectra of green tissues are dominated by the water vibrational absorption, typically presenting overtone bands of the OH bonds at 1450 nm and a combination band at 1940 nm. Partial least square (PLS) regressive computation was also applied both for water and sugar content by performing correlation and predicting models with significant results in terms of correlation and statistical robustness.

Bellincontro et al. (2009) [23] used the same spectral NIR-AOTF device on the wine grape variety Cesanese, also combining it with multiple sensors based on different principles, for non-destructively studying the grape berry response to different conditions of postharvest dehydration. Ten degrees Celsius temperature and two air flow conditions (1.5 and 2.5 m/s, respectively) were proposed as drying parameters, to be compared to 20 °C and 1.5 m/s of ventilation, for observing the impact of environmental factors on the process speed at the same water loss level. The rate of water loss, the modification of the grape berry tissues and, relatively, the consequence on the grape metabolism was also studied. Only the observation of averaged NIR spectra from each withering condition allowed them to discriminate the water loss rate and amount on the basis of the changes observed in terms of water bond absorptions at the characteristic wavelengths, as described before.

Beghi et al. (2015) [115], working on wine grape Corvina, which is the main red colored variety for producing Amarone wine, demonstrated the feasibility of visible near infrared spectroscopy (Vis-NIRs) in measuring some characteristic parameters of withered grapes. Analysis was carried out using a portable device operating between 400 and 1000 nm through the reflectance method of acquisition, which was performed by contact on single grape berry. The wine grapes were dehydrated in a traditional facility for the withering process (*‘Fruttaio’*) according to the commercial protocol of postharvest grape withering often adopted in the Valpolicella area [116]. The initial temperature was of 16 °C and moved down during the process to 9 °C, while the RH ranged between 65 and 70%. Recorded Vis-NIR spectra were destined to chemometric calculations for the building up of qualitative (PCA) and quantitative (PLS) algorithms. PCA allowed them to differentiate the progressive timing of water loss measured on grape berries (weight loss stages), while PLS regressive models were performed with the goal of predicting their sugar content (soluble solids) and firmness. This latter was, in parallel, evaluated via penetrometric technique by using a dynamometer. The main final results reported R^2^ equal to 0.62 and 0.56 and RPD (ratio of performance to deviation) [110] of 1.87 and 1.79 for sugars and firmness, respectively. Reduced robustness of the algorithms, compared to those obtained by other works, were justified by the authors with the limited spectral gain of the employed device. At the same time, they correctly underlined the limited operative costs and the instrumental aptitude to be commercially used.

### 5.2. E-Nose

Typically, an E-Nose is a combination of sensors able to detect gases and volatiles emitted by a product or a matrix. The analytical response of this kind of analytical device is related to the sensor sensitivity and to the sensor array aptitude to discriminate specific volatile or aromatic patterns [117]. The relative low cost of sensors, as well as the reduced time of sample preparation for the analysis compared to the gas-chromatographic techniques, indicates the E-Nose application as a non-destructive technique. During the last years, the use of E-Noses for food evaluation, in relation to quality definition and monitoring of production processes or storage conditions, has been a focus [118].

Santonico et al. (2010) [45] employed, for the first time, an E-Nose based on quartz microbalance (QMB) sensors coupled to metalloporphyrin coatings [119] as a tool for monitoring the aromatic pattern of wine grapes during the dehydration process to which they were subjected. On the Cesanese variety of red wine grape, a test of the dehydration treatment was carried out under controlled conditions at two distinct temperatures, 10 and 20 °C, respectively, keeping the same level of RH (45%) and ventilation (1–1.5 m/s). In correspondence to four stages of grape bunch weight loss (10%, 20%, 30% and 40%), grapes were sampled, and the obtained musts were analyzed, contextually, through the E-Nose and a GC-MS. PCA calculation performed on sensor detections revealed how grape dehydrated samples were clearly grouped based on the temperature process. A quite evident grouping among the four stages of grape weight loss related to the 20 °C of temperature was observed, with the extreme condition of 40% well segregated from the others. Conversely, less strong differences were observed in the samples dehydrated at 10 °C, and their progressive behavior associated to grape water loss revealed a less evident clusterization of grape samples belonging to the four stages of drying. Those observations confirm the already known influence of the temperature on the dehydration process [4,10,30], and also in influencing the aromatic profile of dehydrated grapes. That observation was also corroborated by the GC-MS analysis, which defined specific VOC markers related to metabolic pathways induced by the water loss conditions, such as C6 aldehydes and alcohols, or ethanol, acetaldehyde, acetic acid, and ethyl acetate, already described by other authors [7,20]. The ability of the E-Nose in non-destructively measuring aromatic grape patterns was underlined.

Lopez de Lerma et al. (2012) [48] employed the same QMB E-Nose in the south of Spain (Andalucia) with the purpose of monitoring the aromatic evolution of Pedro Ximénez grapes subjected to a sun-drying process. The timing of grape drying was very short, and the effect was really strong, considering that, in a total of 9 days of exposure, the grape samples lost about 30% of the initial weight, while the sugars concentrated up to 43% starting from the initial 22%. In correspondence to an initial time (0) and 2, 4, 6, and 9 days of sampling (final time), the musts of dehydrated grapes were measured by the E-Nose for aromatic pattern definition, while specific volatiles were detected via GC-MS. VOCs were grouped into aromatic families and also associated to aromatic descriptors and nuances (e.g., ripe fruit, herbaceous, floral, fatty, milky, and toasty) describing the sensorial and organoleptic modification occurring in the grape samples as a consequence of the drying conditions. Applying a multivariate approach of computing, the authors obtained a proper grouping of grape samples on the basis of time of drying, also defining the right moment for ending the process in order to drive the correct aromatic characters of grapes for producing Pedro Ximénez sweet Passito wine.

### 5.3. Chlorophyll Fluorescence

The chlorophyll fluorescence is widely assumed to be a performant technique addressed to detect environmental, chemical, and biological stress in plant cells [120,121,122]. In adopted measurements of chlorophyll fluorescence, a decrease in dark-adapted Fv/Fm and an increase in F0 for describing the presence of environmental responses to several abiotic stresses, such as low or high temperature, strong radiative exposure, and water stress, have been observed [123,124]. The employment of chlorophyll fluorescence techniques has been also addressed to study the postharvest life of fruit and vegetable crops for monitoring several physiological modifications, mainly due to storage conditions and/or postharvest stresses of vegetal tissues [125,126,127]. HarvestWatch system (Isolcell, Laives, Bozen, Italy) is a device based on optical sensors able to measure chlorophyll fluorescence (Fv) that several studies, and following commercial application, have developed to determine low oxygen tolerance limits in horticultural products (e.g., apple) [123,124]. In 2008, for the first time, Ramin et al. (2008) [128] tested the HarvestWatch with the purpose to monitor the water loss in grapes by the evaluation of Fv as indicator of the grape cell stress under dehydration conditions. Thompson and Flame seedless table grapes, usually dried for producing raisins, were dehydrated at a temperature of 20 °C under the effect of ventilation, while the control grape samples were kept at 0 °C in a cold room. The evolution of grape water loss in test samples was monitored together with other technological parameters (e.g., sugars, titratable acidity, pH, and color), while the chlorophyll fluorescence was in-continuous recorded as F**α** values (fluorescence at irradiance F_0_). In 6 days, grapes reached a global water loss amount of about 20% and a very significant correlation was found between the progressive stages of water loss (weight loss) and the F**α** declination. Bellincontro et al. (2009) [23] used the same technological device for monitoring a dehydration process in Cesanese wine grapes destined to produce Passito wine. As already mentioned above, this work focused on the comparison of two distinct conditions of temperatures (10 and 20 °C) and air flow (1.5 and 2.5 m/s) for studying the effects of the intensity and length of the dehydration process on the quality attributes of wine grapes. The different evolution of two postharvest water loss protocols was well monitored through the F**α** trends associated to each specific drying condition. Finally, chlorophyll fluorescence results were correlated to other non-destructive detections (e.g., NIRs) and chemical and biochemical data.

### 5.4. NMR

In recent years, nuclear magnetic resonance and magnetic resonance for imaging (MRI) instruments, even in the low-field (LF-NMR) configuration, have been studied as tools able to perform interesting evaluations in the field of food science and technologies and, in particular, in the quality measurements of fruits and vegetables [129]. Within the so-called non-destructive technologies, magnetic resonance, in its different operative arrangements, has been used for detecting the internal structure and morphology, as well as the water dynamics, of horticultural crops [130]. NMR contributed to elucidate the water-related characteristics, carbohydrates and protein, and other metabolites related to the nutritional quality in food products [131]. Even the internal characteristics from the physical, structural, physiological, and pathological point of view have been well associated to the NMR techniques and its applications [129]. In grapes, MRI has been used for evaluating the internal quality attributes and monitoring the berry ripening evolution [132]. In kiwifruit, Burdon and Clarck (2001) [133] tested the feasibility of MRI technique for detecting fruit water loss. The measurements were based on the aptitude of MR images to describe the fruit volume modification, in turn affected by the structural relaxation due to the water movement. The aqueous environments of the fruit in the deeper internal part (core) and in the pericarp, both for inner and outer sections, as well as the water content associated to the proton density, have been investigated [133]. Bellincontro et al. (2009) [23] included the use of MRI technique in an experimental study where some different sensor-based devices were combined with the aim of non-destructively monitoring the wine grape dehydration under controlled conditions. The authors were able to monitor the texture changes occurring during grape berry dehydration. At the initial time of the drying process and up to 5% of weight loss, no significant differences were detected by the MR images. At 10% of water loss, berries maintained at 20 °C began to show some black areas under the skin tissue as a consequence of the water movement. These dark parts became more and more significant in correspondence to the stage of 15% weight loss, always at 20 °C. The same black spots became visible under the skin of samples dehydrated at 10 °C. At the final stage of 20% of dehydration, berries kept at 20 °C presented quite complete black tissue, while samples dehydrated at 10 °C with 2.5 m/s air flow showed larger areas under the skin of disorganized tissues than the grape berries always treated at 10 °C but with an air flow of 1.5 m/s. The authors inferred that the percentage of grape weight loss, together with the rate of dehydration, have a great influence on the tissue integrity, which is a good parameter to monitor for revealing the water stress, and it can be well detected by the MRI technique.

## 6. Conclusions

Wine grape postharvest water loss is an event associated with grape dehydration strongly affecting the quality of processed grapes and, as a consequence, of the final wines produced. During the last two decades, several authors and research groups have looked at this technological and scientific field with the goal of better elucidating the impact of the water stress occurring to wine grape berries and its effects on metabolite profiles based on the conditions, modulation, length, and intensity of the process. In the present manuscript, a full revision of the scientific papers treating this argument has been reported, starting from an accurate description of the different techniques, naturally and/or artificially managed, that induce the grape water loss. Following that, an overview on the metabolite modification observed when water loss occurs has been included, considering the impact of water stress on metabolite biosynthesis and modulation, as well as their degradation due to the oxidative process linked to senescence occurrence. Secondary metabolites (mainly polyphenols and VOCs) evolution has been also observed as quality attributes potentially affecting the sensorial and organoleptic characters of the final wines. Finally, an excursus on the non-destructive technologies aimed at monitoring the wine grape postharvest water loss has been done, underlining the application of several sensor-based devices for this goal.

## Figures and Tables

**Table 1 metabolites-11-00318-t001:** List of known systems applied to induce grape water loss to produce wine.

Known System of Grape Berry Water Loss	Practice or Technique	Note
Sun-drying	Practice	
Sun-drying in plastic tunnel	Practice/technique	Berry water loss
Uncontrolled dehydration in *fruttaio* (close facility)	Practice/technique	
Controlled dehydration in *fruttaio*	Technique	
Withering in *fruttaio* with ambient control	Technique	Berry water loss and senescence
Cane-cut/twisted on-vine	Practice/technique	Berry water loss and senescence on vine with cut or twisted bunch branch
Late harvest	Practice	Berry water loss and senescence on vine
Noble rot	Practice	*Botrytis cinerea* is the main factor responsible for water loss
Ice wine	Practice	The water loss depends on the length of the freezing process

**Table 2 metabolites-11-00318-t002:** Putative VOC markers of dehydration process in wine grape. CC: controlled conditions; SD: sun-drying.

VOC Chemical Class	Putative Process Markers	Cultivar	Dehydration Protocol	Reference
*Carboxylic acids*	octanoic acid	Pedro Ximeénez	SD	[48]
*Alcohols*	ethanol, Isobutanol	Malvasia, Pedro Ximeénez, Sangiovese, Trebbiano	CC, SD	[20,43,48,49]
*Terpenes*	linalool oxides	Cesanese, Malvasia moscata, Moscato nero d’Acqui	CC	[2,25,29]
*Esters*	ethyl acetate, isoamyl acetate	Malvasia, Sangiovese, Tempranillo, Trebbiano	CC	[30,43]
*Lactones*	γ-valerolactone, γ-butyrolactone	Pedro Ximeénez, Tempranillo	SD	[43,48]
*Phenols*	vinylguaiacol, vanillin	Cesanese, Tempranillo	CC, SD	[25,43]
*Maillard reaction products*	furfural, 5-methylfurfural	Montepulciano, Pedro Ximeénez, Tempranillo	SD	[43,45,48]

**Table 3 metabolites-11-00318-t003:** Putative non-volatile markers of dehydration process in wine grape. CC: controlled conditions; UC: uncontrolled condition.

Non-Volatile Chemical Class	Putative Process Markers	Cultivar	Dehydration Protocol	Reference
*Organic acids*	tartaric acid malic acid	Moscato nero d’Acqui, Amarone	CC	[2,15]
*Anthocyanins*	peonidin-3-*O*-glucoside	Moscato nero d’Acqui, Raboso Piave	CC	[2,26]
*Hydroxybenzoic acid*	protocatechic acid	Xynisteri	UC	[28]
*Hydroxycinnamic acids*	caftaric acid	Xynisteri	UC	[28]
*Flavonols*	quercetin-3-*O*-glucoronide	Raboso Piave, Xynisteri	CC, UC	[26,28]
*Flavan-3-ols*	catechin, epicatechin	Cabernet Sauvignon, Xynisteri	UC	[28,60]
*Stilbenes*	*trans*-resveratrol, *trans*-piceid	Aleatico, Cabernet Sauvignon, Corvina, Raboso Piave, Xynisteri	CC, UC	[26,28,53,60]
*Lignans*	isolariciresinol-β-4′-*O*-glucopyranoside	Xynisteri	UC	[28]

## Data Availability

Not applicable.

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
