# Peer review of "Postharvest Water Loss of Wine Grape: When, What and Why"

_metabolites, 2021, doi:10.3390/metabo11050318_

Round 1

Reviewer 1 Report

The paper "Postharvest water loss of wine grape: when, what and why" describes the main techniques used to induce water loss in wine grapes and gives information on the most important metabolic changes that occur in the berries during water loss. 
The paper is a useful review about the postharvest protocols used in winemaking and about the main biochemical changes induced by these protocols on grapes and wines.
Given the accuracy of the review, I suggest the paper publishing in its current form. 

Author Response

Please find attached a Word file including point-by-point response

Reviewer 2 Report

This review article is focusing on the several different harvests and postharvest practices/techniques of grapes. These practices/techniques affect water loss from the matured berry.

In section #2, the authors introduce several techniques and harvest methods to induce dehydration (drying) of berry. Each technique is briefly and well explained.

In section #3, the effects of water loss from berry on the metabolite changes are well described. VOCs and polyphenols are mainly discussed. Probably, many readers will be interested in this section.

In section #4, the general effects of dehydrated grape berries on wine quality are introduced and discussed. This is an important part to understand the worth of dehydration treatment of grape.

Final section #5 looks like supplemental, however, the technical information will be of some help. The authors concluded non-destructive technology will be useful to monitor the postharvest change of grape berry.

I think this review article provides fundamental knowledge in grape science and helps to understand traditional technology (dehydration, drying treatment of grape) for winemaking. The current version of the manuscript is probably acceptable.

[Very minor revisions]

Table 1, 2, 3>> please align the text in each line to the cell top. In the current version, it is difficult to distinguish the items in the same line.

L.135; Slaghenaufi et al. [17] >> Slaghenaufi et al. (2020)[17]

L.147; Genovese et al. [18] >> Genovese et al. (2007)[18]

L.149; Lukic et al. [19] >> Lukic et al. (2016)[19]

L.164; What is OIV(2018)?  This is not listed in the reference.

L.170; cis-rose oxide >> cis-rose oxide (cis is italic)

L.238; Authors >> authors

L.239; (E) >> (E) (italic)

L.240; Authors >> authors

L.242; (E) >> (E) (italic)

L.245; (E) >> (E) (italic)

L.246; (Z) >> (Z) (italic)

L.251; cis- >> cis- (italic)

L.261; trans- >> trans- (italic)

L.357-361; ****-3-O-**** >> ****-3-O-**** (O is italic)

L.359; Authors >> authors

L.369; p-coumaric >> p-coumaric (p is italic)

L.374; Kaempferol >> kaempferol

L.385; ****-4’-O-**** >> ****-4’-O-**** (O is italic)

L.389; trans-piceid >> trans-piceid (trans is italic)

L.390; E-ε-viniferin, E-astringin and E-miyabenol C >> (E)-ε-viniferin, (E)-astringin and (E)-miyabenol C

L.392; Z-ε-viniferin >> (Z)-ε-viniferin

L.394; E/Z-ε-viniferin >> (E)/(Z)-ε-viniferin ??

L.413; DMR: Double maturation raisonnée >> DMR: double maturation raisonnée

L.442; Authors >> authors

L.455; Zironi and Ferarini [103] >> Zironi and Ferarini (1987)[103]

L.470; Lòpez de Lerma et al [107] >> López de Lerma and Reinado (2011) [107]

L.505; Slaghenaufi et al [17] >> Slaghenaufi et al. (2020)[17]

L.512; (E) >> (E) (italic)

L.527; Moreno et al. [41] >> Moreno et al. (2008)[41]

L.535; Ossola et al. [2] >> Ossola et al. (2017)[2]

L.543; Panceri et al. [13] >> Panceri et al. (2017)[13]

L.549; Mencarelli et al. [10] >> Mencarelli and Bellincontro (2018)[10]

L.554; Clay et al. (2006) >> Clay et al. (2006)[??]  This is not listed in the reference.

L.564; Setkova et al. (2007) >> Setkova et al. (2007)[119]

L.632; Bellincontro et al [126] >> Bellincontro et al. (2011)[126]

L.634; Authors >> authors

L.651; Bellincontro et al. [29] >> Bellincontro et al. (2009)[29]

L.662; Beghi et al. [127] >> Beghi et al. (2015)[127]

L.668; ‘Fruttaio’ >> ‘Fruttaio’ (italic)

L.679; Authors >> authors

L.683; E-nose >> E-Nose

L.691; Santonico et al. [55] >> Santonico et al. (2010)[55]

L.691; E-nose >> E-Nose

L.698; E-nose >> E-Nose

L.710; Authors >> authors

L.711; E-nose >> E-Nose

L.713; E-nose >> E-Nose

L.719; E-nose >> E-Nose

L.724; Authors >> authors

L.741; Ramin et al. [140] >> Ramin et al. (2008)[140]

L.751; Bellincontro et al. [29] >> Bellincontro et al. (2009)[29]

L.773; Burdon and Clarck [145] >> Burdon and Clarck (2001)[145]

L.778; Bellincontro et al. [29] >> Bellincontro et al. (2009)[29]

L.781; Authors >> authors

In Table 3;  

epicatechin >> Epicatechin

 Trans-resveratrol >> trans-Resveratrol

 trans-piceid >> trans-Piceid

Author Response

(The authors gave the same response as above.)

Reviewer 3 Report

The manuscript entitled „Postharvest water loss of wine grape: when, what and why“ is giving an interesting insight into the different practices and methods used to increase wine quality and style, but it needs some changes before being accepted for publishing.

The first thing that I recommend is to clearly separate off-wine (postharvest) grape dehydration from methods used to obtain so-called „prädikat“ wines produced from overripened grapes (a designation based on ripeness level obtained on wine) late harvests, ice wine, and others which are far more complex than presented in this manuscript. Perhaps it would be better only to focus on postharvest dehydration practices (in the vineyard, on the sun, off the sun, and in different levels of control environments).

The second most important thing is related to relative vs. absolute changes during the dehydration process.  In general, since the main purpose of grape dehydration is to increase the content of all metabolites that have a positive impact on resulting wine quality, there are huge differences in the efficiency of different methods, approaches used in obtaining this goal. I considered that this must be incorporated into the manuscript.

It is not correct to state that water loss is inducing sugar accumulation, it is more correct to say that due to the reduction of water content there is a relative increase in sugar content – meaning that all the changes in metabolites must always be compared using relative, but also absolute criteria. Meaning that the same and unchanged absolute amount of TSS (i.e. 200 g of sugars in 1 kg of grape juice before drying can still be the same 200 grams of sugars in 600 grams of grape juice), and the relative change from 20% to  33.33% of TSS is only the consequence of water loss, and not due to sugars accumulation.

These relative/absolute changes must be incorporated in discussion since they are crucial to understanding real changes in metabolites during the dehydration process. For example, if we have an unchanged concentration (relative -%, g/kg, mg/kg) of the specific metabolite in grape juice (or other parts of berry)  it actually means that the absolute amount of this compound decreased more or less linear to water content decrease. So besides the new compounds that are synthesized during the dehydration process, all the others must be compared with the change in water content. The best examples for this are usually the organic acid content changes during dehydration, which are hardly being mentioned in the manuscript.

Author Response

(The authors gave the same response as above.)

Round 2

Reviewer 3 Report

The manuscript entitled "Postharvest water loss of wine grape: when, what and why" modified according to comments and suggestions from the first round of review is now acceptable for publication.